# Posterior Spinal Fusion Surgery for Neuromuscular Disease Patients with Severe Scoliosis Whose Cobb Angle Was over 100 Degrees

**DOI:** 10.3390/medicina59061090

**Published:** 2023-06-05

**Authors:** Masayuki Miyagi, Wataru Saito, Yusuke Mimura, Toshiyuki Nakazawa, Takayuki Imura, Eiki Shirasawa, Kentaro Uchida, Shinsuke Ikeda, Akiyoshi Kuroda, Sho Inoue, Yuji Yokozeki, Yoshihide Tanaka, Tsutomu Akazawa, Masashi Takaso, Gen Inoue

**Affiliations:** 1Department of Orthopaedic Surgery, School of Medicine, Kitasato University, Tokyo 252-0374, Japan; boatwataru0712@gmail.com (W.S.); msm.men.36@gmail.com (Y.M.); nakazawa@kitasato-u.ac.jp (T.N.); tk2003@kitasato-u.ac.jp (T.I.); eeiikkii922@yahoo.co.jp (E.S.); kuchida@med.kitasato-u.ac.jp (K.U.); ikedas@kitasato-u.ac.jp (S.I.); akiyoshikvroda@yahoo.co.jp (A.K.); ino2510@gmail.com (S.I.); yuji0328yoko@yahoo.co.jp (Y.Y.); yo.millennium@gmail.com (Y.T.); mtakaso@kitasato-u.ac.jp (M.T.); ginoue@kitasato-u.ac.jp (G.I.); 2Department of Orthopaedic Surgery, St. Marianna University School of Medicine, Kawasaki 216-8511, Japan; cds00350@par.odn.ne.jp

**Keywords:** neuromuscular diseases, scoliosis, posterior spinal fusion surgery, Cobb angle

## Abstract

*Background and objectives*: Patients with neuromuscular diseases usually have progressive neuromuscular scoliosis (NMS), requiring invasive surgery. Some patients present with severe scoliosis at the time of consultation and are difficult to treat. Posterior spinal fusion (PSF) surgery combined with anterior release and pre- or intraoperative traction would be effective for severe spinal deformities but would be invasive. This study aimed to evaluate the outcomes of PSF-only surgery for patients with severe NMS with a Cobb angle > 100°. *Materials and Methods*: Thirty NMS patients (13 boys and 17 girls; mean age 13.8 years) who underwent PSF-only surgery for scoliosis with a Cobb angle > 100° were included. We reviewed the lower instrumented vertebra (LIV), duration of surgery, blood loss, perioperative complications, preoperative clinical findings, and radiographic findings, including Cobb angle and pelvic obliquity (PO) in the sitting position pre- and postoperatively. The correction rate and correction loss of the Cobb angle and PO were also calculated. *Results***:** The mean duration of surgery was 338 min, intraoperative blood loss was 1440 mL, preoperative %VC was 34.1%, FEV1.0 (%) was 91.5%, and EF was 66.1%. There were eight cases of perioperative complications. The Cobb angle and PO correction rates were 48.5% and 42.0%, respectively. We divided the patients into two groups: the L5 group, in which the LIV was L5, and the pelvis group, in which the LIV was the pelvis. The duration of surgery and PO correction rate in the pelvis group were significantly higher than those in the L5 group. *Conclusions*: Patients with severe NMS demonstrated severe preoperative restrictive ventilatory impairments. PSF surgery without anterior release or any intra-/preoperative traction showed satisfactory outcomes, including acceptable scoliosis correction and improved clinical findings, even in patients with extremely severe NMS. Instrumentation and fusion to the pelvis for severe scoliosis in patients with NMS showed good PO correction and low correction loss of Cobb angle and PO, but a longer duration of surgery.

## 1. Introduction

Patients with neuromuscular diseases such as muscular dystrophy and spinal muscular atrophy usually show progressive scoliosis called neuromuscular scoliosis (NMS). Some patients with NMS exhibit an impaired quality of life, including sitting difficulty, pulmonary dysfunction, and cardiac dysfunction, and usually require surgery [1,2]. Reportedly, posterior spinal correction and fusion surgery for NMS have good correction and clinical outcomes, including inhibition of scoliosis progression and improvement in sitting difficulty [2,3]. However, many patients with NMS show pulmonary and cardiac dysfunction preoperatively, and several authors have reported high perioperative complication rates [4,5]. We also previously reported that 24% of patients with NMS showed perioperative complications, and patients with major perioperative complications exhibited severe restrictive ventilatory impairment preoperatively [6]. Therefore, early intervention, including spinal fusion surgery, is recommended before pulmonary function impairment becomes severe.

However, some patients with NMS show severe scoliosis at the time of consultation and are difficult to treat. For the severe spinal deformity, posterior spinal fusion (PSF) surgery combined with anterior release, intraoperative halo-femoral traction, or preoperative halo-gravity traction would be effective. However, surgery for patients with severe NMS is usually highly invasive, with a long duration of surgery and a high amount of blood loss. There are several concerns about these invasive procedures for patients with fragile conditions. We performed PSF surgery without any anterior release or pre- or intraoperative traction in all NMS patients, even in patients with severe spinal deformities. This study aimed to evaluate the outcomes of PSF-only surgery in patients with severe NMS with a Cobb angle > 100°.

## 2. Materials and Methods

### 2.1. Patient Population

A total of 30 patients (13 boys and 17 girls) who underwent PSF-only surgery for scoliosis with a Cobb angle > 100° from 2006 to 2020 at a minimum 1-year follow-up were included. The mean age at surgery was 13.8 years (range, 9–17 years), and the mean follow-up period was 57.9 months (range, 12–135 months). The diagnoses included 11 patients with spinal muscle atrophy, 8 patients with Duchenne type muscle dystrophy, and 11 patients with other types of atrophy, including Fukuyama type muscle dystrophy, Ulrich type muscle dystrophy, and central core disease. All the patients had a flaccid-type neuromuscular disease and were non-ambulatory. Most patients could communicate without trouble; however, some had mental retardation. All patients underwent preoperative pulmonary function evaluation via spirometry and cardiac function evaluation via echocardiography. All patients experienced sitting difficulty and back pain due to NMS. None of the patients were excluded because of poor preoperative physical status.

### 2.2. Surgical Procedure

All surgeries were performed under general anesthesia. Motor-evoked potentials were used to monitor spinal cord function in all cases. In addition, we performed an autotransfusion as preoperative storage, while an intraoperative collection was performed during surgery. An incision was made in the midline of the back, and the spinal structure was exposed from the upper thoracic spine to the sacrum, or pelvis. After removing all soft tissues, instrumentation was performed using pedicle screws, hooks, and sublaminar cables (Nesplon Cable System; Alfresa, Tokyo, Japan). Initially, fusion levels ranged from T4 to L5 until May 2018. However, starting in June 2018, the determination of the fusion levels was based on individual cases, taking into account factors such as curve flexibility, apex, and physical status. In cases with severe deformity, high pelvic obliquity, and rigid curves, PSF surgery to the pelvis was considered, provided that the patient exhibited adequate physical status. We then obtained local autograft bone from the spinous processes, lower facet joints, and transverse processes. A ponte osteotomy was performed on several segments (usually four or five) around the scoliosis apex to release the spinal structure and obtain curve flexibility. In all cases, spinal deformities were corrected using two combined techniques: the cantilever technique and the rod rotation technique. After correction, all laminae and facet joints were decorticated, and a local autograft bone mixed with a bioresorbable bone graft was placed. Finally, the wound was sutured in three layers, and drainage tubes were placed. In all cases, we did not perform any anterior release surgery or preoperative or intraoperative traction. We attempted extubation as soon as possible after surgery when the patients could ventilate their lungs spontaneously to prevent respiratory dependency.

### 2.3. Measurements

We reviewed age at surgery, preoperative %VC and FEV1.0 (%) for pulmonary function, preoperative ejection fraction (EF) for cardiac function, duration of surgery, blood loss, perioperative complications, upper instrumented vertebra (UIV), lower instrumented vertebra (LIV), and radiographic findings including Cobb angle and pelvic obliquity (PO) in the sitting position preoperatively, 1 month postoperatively, and at the final follow-up. The correction rates for Cobb angle and PO were calculated as follows:

Correction rate (%) = (preoperative angle − 1-month postoperative angle)/preoperative angle × 100.

In addition, the correction losses for the Cobb angle and PO were calculated as follows:

Correction loss (degrees) = angle at final follow-up − 1-month postoperative angle.

### 2.4. Statistics

A repeated measures analysis of variance (ANOVA) was performed to compare the radiographic findings preoperatively, 1 month postoperatively, and at the final follow-up. Post hoc analysis was performed using Tukey’s test for multiple comparisons. Furthermore, we divided the patients into two groups: the LIV was an L5 group (L5), and the LIV was the pelvis group (pelvis). Age at surgery, preoperative %VC and FEV1.0 (%), preoperative EF, duration of surgery, blood loss, and radiographic findings, including Cobb angle and PO, were compared between the two groups. Leven’s test was used to assess the equality of the variance of the variables of interest. The Mann-Whitney U test was applied to variables with unequal variances. For variables with equal variances, an unpaired *t*-test was used. Statistical significance was set at *p* < 0.05.

### 2.5. Ethics

Ethical approval from our institutional review board (IRB) was obtained for this study. This study was conducted in accordance with the ethical principles specified in the 1964 Declaration of Helsinki and its later amendments.

## 3. Results

The patient characteristics and surgical outcomes are described in Table 1. The mean preoperative %VC was 34.1%, FEV1.0 was 91.5%, and EF was 66.1%. Preoperative %VC in patients with severe NMS was extremely low, although preoperative FEV1.0 and EF were maintained. All patients reported difficulty sitting and back pain due to severe scoliosis preoperatively. For surgery, the UIV was T3 in 1 case, T4 in 25 cases, and T5 in four cases. The LIV was L5 in 24 cases and the pelvis in 6 cases. The preoperative and postoperative mean Cobb angles were 121.9°and 63.8°, respectively, and the mean correction rate was 48.5%. The preoperative and postoperative mean PO were 42.9° and 25.5°, respectively, and the mean correction rate was 42.0%. In addition, the duration of surgery was 338.6 min, and blood loss was 1441.1 mL, indicating the need for a highly invasive surgery (Table 1). Postoperatively, all patients showed improvement in sitting and back pain. There were eight cases (24%) with perioperative complications, including pneumonia, CO_2_ narcosis, urinary tract infection, hemodynamic instability, and surgical site infection. Four of these eight cases had respiratory complications (Table 2). At the final follow-up, no cases required revision surgery, and all cases demonstrated successful maintenance of correction as well as improvements in sitting difficulty and back pain.

When we divided the patients into two groups, the L5 group and the pelvis group, the duration of surgery and PO correction rate in the pelvis group were significantly higher than those in the L5 group. Furthermore, the correction loss of the Cobb angle and PO in the pelvis group was significantly lower than in the L5 group. However, there were no significant differences in age at surgery, preoperative %VC, preoperative FEV1.0, preoperative EF, preoperative Cobb angle, preoperative PO, blood loss, or correction rate of the Cobb angle between both groups (Table 3; Figure 1).

### Representative Case Presentation

Patient: A 13-year-old girl was diagnosed with central core disease secondary to motor retardation at the age of 3 years. Subsequently, at 10 years old, she began experiencing symptoms of spinal scoliosis, which gradually worsened over time. After turning 12, she began experiencing difficulty sitting and back pain, leading to her referral to our institution for spinal scoliosis treatment. A spinal X-ray from the frontal (Figure 2A) and lateral (Figure 2B) views while the patient was in a sitting position revealed severe spinal scoliosis, with a Cobb angle measuring 115.6° and a PO measuring 40.5°. In addition, a spinal X-ray in a supine position under traction revealed spinal scoliosis with a Cobb angle correction of 39.1% (Figure 2C). Under general anesthesia, we performed corrective spine surgery using posterolateral fusion instrumented from T4 to the pelvis. The procedure lasted for 7 h and 31 min, with a blood loss of 1520 mL. Postoperative spinal X-ray revealed successful spinal correction with a Cobb angle measuring 52.1° and a 54.9% correction and a PO measuring 12.0° and a 70.4% correction (Figure 3A,B). The patient was discharged from the intensive care unit after 2 days and started physical therapy 3 days following the surgery. Gradually, her sitting difficulty improved, and she was discharged from the hospital 25 days postoperatively. At the final follow-up 2 years after the surgery, the patient successfully maintained correction and experienced improvements in sitting difficulty and back pain (Figure 4A,B).

## 4. Discussion

In this study, patients with severe NMS had extremely low preoperative %VC. PSF surgery for severe NMS showed satisfactory outcomes, although it required highly invasive surgery with a high complication rate. PSF surgery to the pelvis showed a longer duration of surgery, a higher PO correction rate, and a lower correction loss of Cobb angle and PO compared to fusion surgery to L5.

Regarding the characteristics of patients with severe NMS in this study, patients with severe NMS with a Cobb angle > 100° showed extremely low %VC preoperatively. Several studies on idiopathic scoliosis patients have reported that progressing scoliosis and increasing Cobb angle were associated with pulmonary dysfunction, especially reduced %VC [7,8]. Furthermore, several studies on neuromuscular diseases have reported that patients with scoliosis show low pulmonary function, especially restrictive ventilatory impairment [6,9,10]. In their review of NMS patients, Mayer et al. reported altered respiratory mechanics due to scoliosis and decreased muscle strength due to neuromuscular diseases, leading to severe pulmonary dysfunction [10]. These findings indicated that patients with severe NMS showed severe restrictive ventilatory impairment.

Regarding the surgery for severe NMS, patients with severe NMS required highly invasive surgery with a longer duration of surgery and a high amount of blood loss, and the perioperative complication rate was 24% in this study. Moreover, many perioperative complications are also associated with pulmonary complications. Toombs et al. reported the following in their study of idiopathic scoliosis patients: Increased spinal curve magnitude might lead to a longer duration of surgery and more perioperative complications [11]. No reports have elucidated the relationship between spinal curve magnitude and the invasiveness of surgery in patients with NMS. However, patients with NMS reportedly show pulmonary dysfunction, which might be a risk factor for NMS surgery [6,11,12]. In addition, several authors have reported that the perioperative complication rate of NMS surgery was 24–68% [4,5,6,13,14], similar to the results of this study. Previous reports also indicated that most perioperative complications are associated with respiration [4,6,13], suggesting that patients with severe NMS require highly invasive surgery with a high complication rate. In particular, when treating patients with severe NMS, attention should be paid to perioperative respiratory complications.

Regarding the surgical strategy for severe NMS, anterior release and posterior fusion surgery, intraoperative halo-femoral traction, and preoperative halo-gravity traction have been reported to be effective in severe scoliosis surgery, including NMS and idiopathic scoliosis [15,16,17,18,19,20,21,22,23]. Auerbach et al. reported that good PO correction in patients with NMS with larger and less flexible curves could be achieved using anterior release and posterior fusion surgery [21]. In contrast, several authors have reported that PSF surgery with intraoperative halo-femoral traction for scoliosis with spastic neuromuscular diseases, including cerebral palsy, was less invasive, including a shorter operation time, a smaller amount of blood loss, and a lower frequency of respiratory complications than PSF surgery with anterior release [22,23]. According to these previous reports, PSF surgery with intraoperative halo-femoral traction for severe NMS is superior in efficacy and safety. However, there were concerns that intraoperative traction induces neuromonitoring signal changes and neurological deficits [24]. Furthermore, in this study, PSF surgery without anterior release or intraoperative traction achieved acceptable scoliosis correction and improved clinical findings. Therefore, considering the results of this study, PSF-only surgery for severe NMS might be sufficient.

Regarding the differences between NMS and adolescent idiopathic scoliosis (AIS), Faldini C. et al. demonstrated a good correction rate (65.0%) of the main curve in AIS patients with Cobb angles exceeding 90° using PSF surgery alone [25]. Additionally, Traversari M. et al. reported a mean correction rate of 58.6% for the major curve in severe AIS patients undergoing PSF surgery through a one-stage posterior-only approach, as shown in their systematic review and meta-analysis [26]. In the present study targeting severe NMS patients, we achieved a significant correction of the Cobb angle (48.5%), albeit lower than the previously reported numbers in severe AIS patients. Maximized correction of scoliosis in severe AIS patients has been associated with high pedicle screw density and aggressive posterior osteotomy techniques [26]. In contrast, for severe NMS patients, there were several concerns about utilizing the high pedicle screw density and aggressive osteotomy techniques owing to the small skeletal structure and poor physical status. Therefore, these considerations may contribute to the comparatively lower correction rates observed in NMS patients compared to AIS patients. However, the main aim of the NMS surgery was to improve clinical findings, including sitting difficulty and back pain, unlike AIS surgery. In this case series, we observed improvements in clinical findings in all cases, suggesting that PSF surgery alone for severe NMS may be sufficient, even in patients with severe NMS.

For the fusion level of NMS surgery in this study, fusion surgery to the pelvis showed a longer duration of surgery, a higher PO correction rate, and a lower correction loss of Cobb angle and PO than fusion surgery to L5. We previously reported that fusion surgery to L5 for NMS showed satisfactory outcomes, including good correction and a high level of safety [27,28]. However, there were some limitations in ending instrumentation for NMS surgery at L5. We also reported that patients with a larger preoperative Cobb angle might not achieve adequate spinal and pelvic correction by fusion only to L5 [29]. Tøndevold reported that pelvic fixation for NMS surgery showed good correction and decreased the risk of correction loss [30]. According to these previous reports, pelvic fixation surgery would be superior in terms of good correction and low correction loss in patients with severe NMS. However, there were some concerns regarding pelvic fixation from a safety standpoint. Hyun et al. reported that pelvic fixation requires extensive surgical exposure, which is associated with a high risk of blood loss and infection [31]. These findings indicate that fixation to the pelvis in patients with severe NMS appears to have a longer operation duration, although good PO correction and low correction loss were observed. Furthermore, when treating patients with severe NMS, there is a need to balance efficacy and safety depending on patient characteristics, including preoperative pulmonary function, curve spinal flexibility, and neuromuscular disease (flaccid or spastic type). In addition, we considered an early intervention for NMS before it became a severe deformity requiring highly invasive surgery.

The study had some limitations. First, only a PSF surgery was performed. We did not have any data on findings from combined anterior release surgery or interoperative traction. Second, we could not evaluate the clinical findings using any questionnaire, although all patients showed improvement in sitting difficulty and back pain. Further studies are needed to evaluate the quality of life of these patients. Third, which is a critical issue, most previous reports of NMS targeted patients with spastic neuromuscular diseases, including cerebral palsy, or were included in each type of neuromuscular disease, whereas all the patients in this study had the flaccid-type neuromuscular disease. There might be some differences in surgical strategy and clinical outcomes between the spastic and flaccid types. Further studies targeting patients with only flaccid-type neuromuscular diseases are needed. Finally, it is important to acknowledge the presence of selection biases in PSF surgery involving the L5 or pelvis. The determination of fusion levels was influenced by various factors, including the patient’s physical status, curve flexibility, and apex level. Therefore, the comparisons between the L5 and pelvis groups inherently incorporated these selection biases. To obtain more accurate results, future studies with no bias will be necessary.

## 5. Conclusions

Patients with severe NMS demonstrated severe preoperative restrictive ventilatory impairments. PSF surgery without anterior release or any intra-/preoperative traction showed satisfactory outcomes, including acceptable scoliosis correction and improved clinical findings, even in patients with extremely severe NMS. Instrumentation and fusion to the pelvis for severe scoliosis in patients with NMS showed good PO correction and low correction loss of Cobb angle and PO, but a longer duration of surgery.

## Figures and Tables

**Figure 1 medicina-59-01090-f001:**
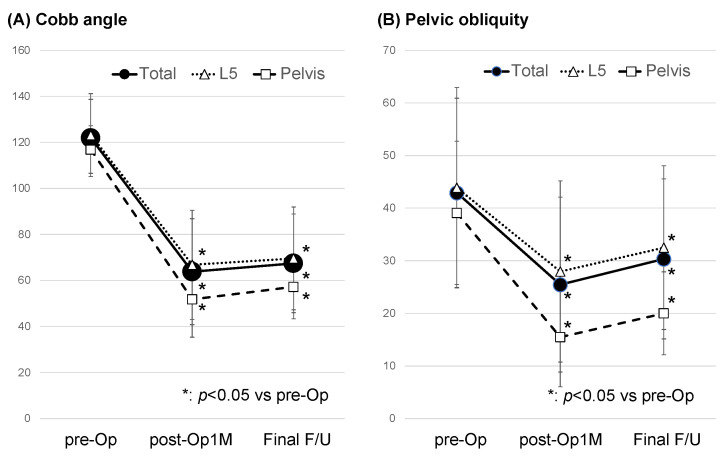
(**A**) Cobb angle and (**B**) pelvic obliquity preoperatively, 1 month postoperatively, and at the final follow-up in whole cases, L5 groups, or pelvis groups. pre-Op: preoperative; post-Op 1M: 1 month postoperative; Final F/U: at final follow-up.

**Figure 2 medicina-59-01090-f002:**
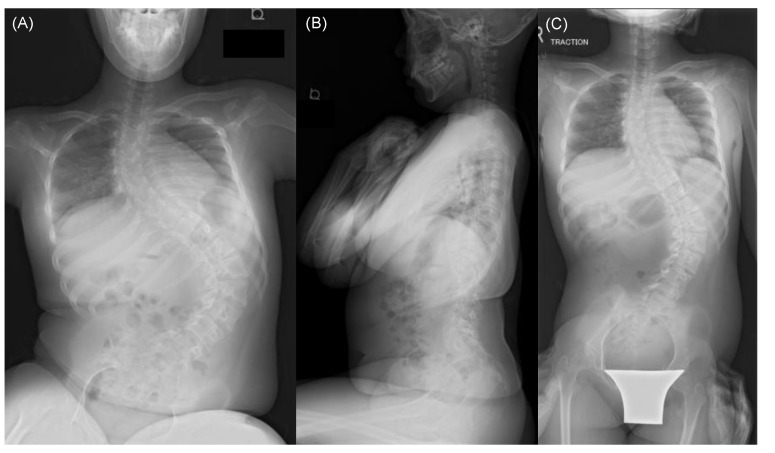
Preoperative X-ray image of the spine from a frontal (**A**) and lateral (**B**) view at sitting and a frontal view at supine under traction (**C**) in a representative case.

**Figure 3 medicina-59-01090-f003:**
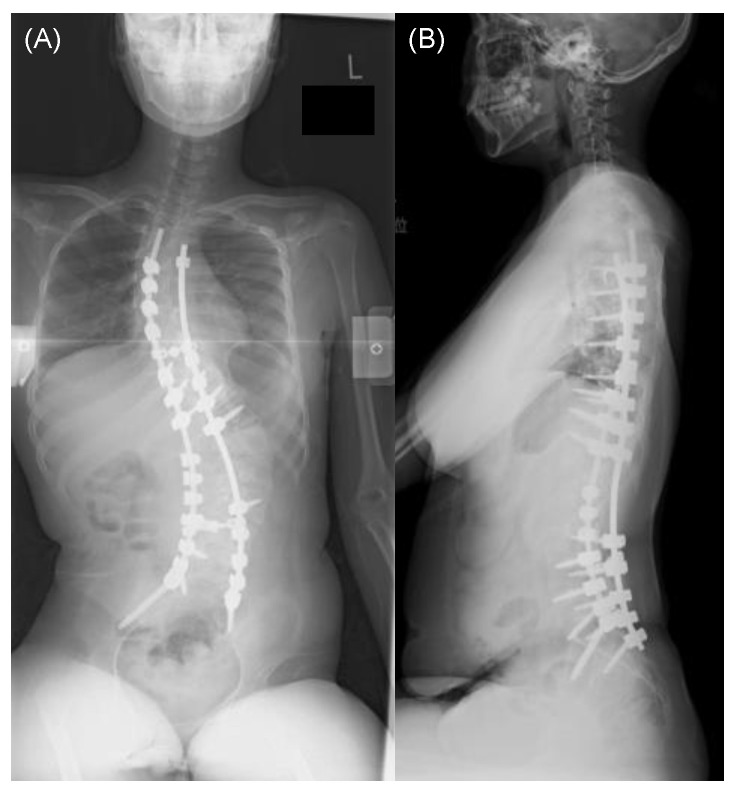
Postoperative X-ray image of the spine from a frontal (**A**) and lateral (**B**) view while sitting in a representative case.

**Figure 4 medicina-59-01090-f004:**
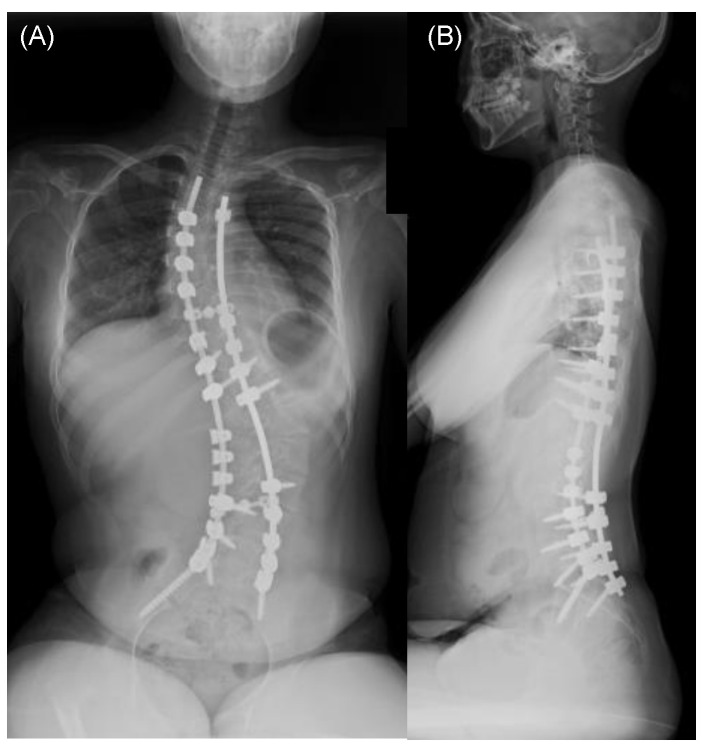
Final follow-up X-ray image of the spine from a frontal (**A**) and lateral (**B**) view while sitting in a representative case.

**Table 1 medicina-59-01090-t001:** Patient characteristics and surgical outcomes.

	Mean	SD
age (years)	13.833	2.306
%VC (%)	34.108	20.535
FEV1.0 (%)	91.524	6.780
EF (%)	66.089	10.066
pre-Op Cobb (°)	121.910	16.792
pre-Op PO (°)	42.920	17.990
Duration of surgery (min)	338.600	74.289
Blood loss (mL)	1441.100	782.407
post-Op1M Cobb (°)	63.840	23.008
post-Op1M PO (°)	25.463	16.615
CR of Cobb (%)	48.513	13.404
CR of PO (%)	41.956	25.123
CL of Cobb (°)	4.430	7.311
CL of PO (°)	4.947	6.388
F/U duration (M)	57.867	40.956

Pre-Op: preoperative; post-Op 1M: 1 month postoperative; CR: correction rate; CL: correction loss; final F/U (M): at final follow-up (months); EF: ejection fraction; PO: pelvic obliquity.

**Table 2 medicina-59-01090-t002:** Perioperative complications.

Complications	Total	L5	Pelvis
	8	6	2
Pneumonia	3	2	1
CO_2_ narcosis	1	1	
Urinary tract infection	2	1	1
Hemodynamically unstable	1	1	
surgical site infection	1	1	

**Table 3 medicina-59-01090-t003:** Comparisons of patient characteristics and surgical outcomes between the L5 group and the pelvis group.

	L5		Pelvis		*p*-Value
	Mean	SD	Mean	SD	
age (years)	14.167	2.390	12.500	1.378	0.115
%VC (%)	31.904	19.386	51.333	25.007	0.125
FEV1.0 (%)	92.391	6.305	85.000	8.718	0.078
EF (%)	64.238	9.985	72.333	9.985	0.081
pre-Op Cobb (°)	123.167	18.004	116.883	10.319	0.422
pre-Op PO (°)	43.875	19.051	39.100	13.623	0.570
**Duration of surgery (min)**	**323.042**	**71.289**	**400.833**	**53.124**	**0.019**
Blood loss (mL)	1558.583	820.464	971.167	350.328	0.101
post-Op1M Cobb (°)	66.846	23.699	51.817	16.427	0.156
post-Op1M PO (°)	27.958	17.211	15.483	9.436	0.101
CR of Cobb (%)	46.736	12.997	55.621	13.758	0.150
**CR of PO (%)**	**37.265**	**24.272**	**60.720**	**20.557**	**0.038**
**CL of Cobb (°)**	**5.275**	**7.968**	**1.050**	**0.903**	**0.013**
**CL of PO (°)**	**5.763**	**6.891**	**1.683**	**1.507**	**0.018**
**F/U duration (M)**	**68.375**	**39.097**	**15.833**	**6.882**	**0.003**

Pre-Op: preoperative; post-Op 1M: 1 month postoperative; CR: correction rate; CL: correction loss; final F/U (M): at final follow-up (months).

## Data Availability

The data presented in this study are available on request from the corresponding author.

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
