# Peer review of "Posterior Spinal Fusion Surgery for Neuromuscular Disease Patients with Severe Scoliosis Whose Cobb Angle Was over 100 Degrees"

_medicina, 2023, doi:10.3390/medicina59061090_

Round 1

Reviewer 1 Report

Thanks to the authors for putting their efforts in the present manuscript. 

This paper sought to expose the results of posterior spinal fusion in severe neuromuscolar scoliosis. Despite the importance of the topic and the good architecture of the present paper, this reviewer suggests acceptance after major revisions.

The key problems in this reviewer's opinion are the following:

In a original surgical paper like the present one, there must be present some clinical cases. There are no case presented in this paper, I thinks that's a must for acceptance.

Moreover, this paper does not really add something to the literature on this topic. What is the novelty of the present paper? what a surgeon could benefit in his clinical management? I'd try to extrapolate the novelty from your series 

In addition, how in the present series are not any revision surgery? did you experienced one? 

Thanks you for your effort 

Author Response

Reviewer #1:

  1. In an original surgical paper like the present one, there must be present some clinical cases. There are no case presented in this paper, I thinks that's a must for acceptance.

Response to reviewer’s question #1-1

We sincerely appreciate the effort put into reviewing our manuscript, and we fully agree with your suggestions. Following the reviewer's recommendation, we have included a representative case presentation as follows:

Representative Case Presentation

‘Patient: A 13-year-old girl was diagnosed with central core disease secondary to motor retardation at the age of 3 years. Subsequently, at 10 years old, she began experiencing symptoms of spinal scoliosis, which gradually worsened over time. After turning 12, she began experiencing difficulty sitting and back pain, leading to her referral to our institution for spinal scoliosis treatment. A spinal X-ray from the frontal (Figure 2A) and lateral (Figure 2B) view while the patient was in a sitting position revealed severe spinal scoliosis, with a Cobb angle measuring 115.6° and a PO measuring 40.5°. In addition, a spinal X-ray in a supine position under traction revealed spinal scoliosis with a Cobb angle correction of 39.1% (Figure 2C). Under general anaesthesia, we performed corrective spine surgery using posterolateral fusion instrumented from T4 to the pelvis. The procedure lasted for 7 h 31 min, with a blood loss of 1520 mL. Postoperative spinal X-ray revealed successful spinal correction with a Cobb angle measuring 52.1° and a 54.9% correction and a PO measuring 12.0° and a 70.4% correction (Figures 3A and 3B). The patient was discharged from the intensive care unit after 2 days and started physical therapy 3 days following the surgery. Gradually, her sitting difficulty improved, and she was discharged from the hospital 25 days postoperatively. At final the follow-up 2 years after the surgery, the patient successfully maintained correction and experienced improvements in sitting difficulty and back pain (Figures 4A, B).’

Figure 2. Pre-operative X-ray image of the spine from a frontal (A) and lateral (B) view at sitting and frontal view at supine under traction (C) in a representative case

Figure 3. Post-operative X-ray image of the spine from a frontal (A) and lateral (B) view at sitting in a representative case

Figure 4. Final follow-up X-ray image of the spine from a frontal (A) and lateral (B) view at sitting in a representative case

  1. Moreover, this paper does not really add something to the literature on this topic. What is the novelty of the present paper? what a surgeon could benefit in his clinical management? I'd try to extrapolate the novelty from your series.

Response to reviewer’s question #1-2

We appreciate the reviewer’s feedback and insight. As mentioned in the third paragraph of the discussion section, the novelty of our study lies in demonstrating that PSF surgery without anterior release or intra- /preoperative traction can achieve acceptable scoliosis correction and improved clinical outcomes, even in patients with extremely severe scoliosis. However, we acknowledge that our statements may not have been clear to readers. Therefore, we have revised the following sentences in the abstract and conclusion sections:

‘PSF surgery without anterior release or any intra- /preoperative traction showed satisfactory outcomes, including acceptable scoliosis correction and improved clinical findings, even in patients with extremely severe NMS.’

  1. In addition, how in the present series are not any revision surgery? did you experienced one?

Response to reviewer’s question #1-3

We thank the reviewer’s question. We are pleased to report that no revision surgery was necessary at the final follow-up. Consequently, we believe that PSF surgery without anterior release or intra- /preoperative traction to be a safe approach, which can be considered a strength of this case series. As per the reviewer's suggestion, we have included more information about the absence of revision surgery in the results section, as follows:

‘At the final follow-up, no cases required revision surgery, and all cases demonstrated successful maintenance of correction, as well as improvements in sitting difficulty and back pain.’

Reviewer 2 Report

Thank you for the opportunity to review this manuscript. The authors evaluate the outcome of posterior spinal fusion (without anterior surgery) for the treatment of severe neuromuscular scoliosis. I found the article interesting and well written. 

I have some minor concerns

- I would like you to clarify the criteria for choosing the fusion area: when L5? when pelvis? did pelvic obliquity play a role in this choice?

- was any of you patients able to walk before surgery? did this play a role in deciding whether or not to include the pelvis? 

- as for the discussion, I would like you to better address the differences between idiopathic and neuromuscular scoliosis; I think two recent studies on the topic are interesting and would be worth mentioning:  Traversari M et al, Surgical treatment of severe adolescent idiopathic scoliosis through one-stage posterior-only approach: A systematic review and meta-analysis - doi 10.4103/jcvjs.jcvjs_80_22 , and Faldini C et al One stage correction via the Hi-PoAD technique for the management of severe, stiff, adolescent idiopathic scoliosis curves > 90° - doi 10.1007/s43390-023-00663-4

English language is of good quality 

Author Response

Reviewer #2:

  1. I would like you to clarify the criteria for choosing the fusion area: when L5? when pelvis? did pelvic obliquity play a role in this choice?

Response to reviewer’s question #2-1

We sincerely appreciate the reviewer’s diligent review of our manuscript and raising this question. Initially, our PSF surgery for NMS cases was performed up to the L5 level. However, starting from June, 2018, we began considering PSF surgery to the pelvis in select cases based on several factors, including curve flexibility, apex, and physical status. As per the reviewer's comment, we also recognized that high pelvic obliquity played a significant role in our decision to perform PSF surgery to the pelvis. We have incorporated the following sentences into the surgical procedure section:

‘Initially, fusion levels ranged from T4 to L5 until May, 2018. However, starting from June, 2018, the determination of the fusion levels was based on individual cases, taking into account factors, such as curve flexibility, apex, and physical status. In cases with severe deformity, high pelvic obliquity, and rigid curves, PSF surgery to the pelvis was considered, provided that the patient exhibited adequate physical status.’

This study also had some limitations, particularly in terms of selection biases in choosing between PSF surgery to the L5 or pelvis. Thus, we have included the following sentences in the limitations section:

‘Finally, it is important to acknowledge the presence of selection biases in PSF surgery involving the L5 or pelvis. The determination of fusion level was influenced by various factors, including the patient's physical status, curve flexibility, and apex level. Therefore, the comparisons between the L5 and pelvis groups inherently incorporated these selection biases. To obtain more accurate results, future studies with no bias will be necessary.’

  1. was any of you patients able to walk before surgery? Did this play a role in deciding whether or not to include the pelvis?

Response to reviewer’s question #2-2

Thank you for bringing this to our attention. In the current study, ambulatory individuals were not included, and therefore their ambulatory statuses did not influence the decision of fusion level. To address this point in the manuscript, we have added the following sentences to the ‘Patient population’ section:

‘All the patients had a flaccid-type neuromuscular disease and were non-ambulatory.’

  1. as for the discussion, I would like you to better address the differences between idiopathic and neuromuscular scoliosis; I think two recent studies on the topic are interesting and would be worth mentioning: Traversari M et al, Surgical treatment of severe adolescent idiopathic scoliosis through one-stage posterior-only approach: A systematic review and meta-analysis - doi 10.4103/jcvjs.jcvjs_80_22 , and Faldini C et al One stage correction via the Hi-PoAD technique for the management of severe, stiff, adolescent idiopathic scoliosis curves > 90° - doi 10.1007/s43390-023-00663-4

Response to reviewer’s question #2-3

We appreciate that the two papers the reviewer mentioned have been helpful in strengthening our study. We fully agree that addressing the differences between NMS and AIS is crucial in our discussion. Following the reviewer’s suggestion, we have added the following sentences in the discussion section and its references:

‘Regarding the differences between NMS and adolescent idiopathic scoliosis (AIS), Faldini C et al. demonstrated a good correction rate (65.0%) of the main curve in AIS patients with Cobb angles exceeding 90° using PSF surgery alone [25]. Additionally, Traversari M et al. reported a mean correction rate of 58.6% for the major curve in severe AIS patients un-dergoing PSF surgery through a one-stage posterior-only approach, as shown in their systematic review and meta-analysis [26]. In the present study targeting severe NMS patients, we achieved a significant correction of the Cobb angle (48.5%), albeit lower than the previously reported numbers in severe AIS patients. Maximized correction of scoliosis in severe AIS patients has been associated with high pedicle screw density and aggressive posterior osteotomy techniques [26]. In contrast, for severe NMS patients, there were several concerns about utilizing the high pedicle screw density and aggressive osteotomy techniques owing to the small skeletal structure and poor physical status. Therefore, these considerations may contribute to the comparatively lower correction rates observed in NMS patients compared to the AIS patients. However, the main aim of the NMS surgery was to improve clinical findings, including sitting difficulty and back pain, unlike AIS surgery. In this case series, we observed improvements in clinical findings in all cases, suggesting that PSF surgery alone for severe NMS may be sufficient, even in patients with severe NMS.’

  1. Faldini, C.; Viroli, G.; Barile, F.; Manzetti, M.; Ialuna, M.; Traversari, M.; Vita, F.; Ruffilli, A. One stage correction via the Hi-PoAD technique for the management of severe, stiff, adolescent idiopathic scoliosis curves > 90°. Spine Deform 2023, doi:10.1007/s43390-023-00663-4.
  2. Traversari, M.; Ruffilli, A.; Barile, F.; Viroli, G.; Manzetti, M.; Vita, F.; Faldini, C. Surgical treatment of severe adolescent idiopathic scoliosis through one-stage posterior-only approach: A systematic review and meta-analysis. J Craniovertebr Junction Spine 2022, 13, 390-400, doi:10.4103/jcvjs.jcvjs_80_22.

Round 2

Reviewer 1 Report

The authors followed the reviewers comments accordingly. 
I recomend acceptance in its present form

Kind regards